Identification and validation of prognostic and tumor microenvironment characteristics of necroptosis index and BIRC3 in clear cell renal cell carcinoma

Wei Kai 1
Zhang Xi 2
Yang Dongrong 1 doc_ydr@163.com
1 Urology, The Second Affiliated Hospital of Soochow University , Suzhou, Jiangsu , China
2 Urology, The State Key Lab of Reproductive; The First Affiliated Hospital of Nanjing Medical University , Nanjing, Jiangsu , China
Bamodu Oluwaseun Adebayo
Electronic publication date: 2023 Dec 18
Publication date: 2023
Volume: 11
Electronic Location ID: e16643
Received 2023 Jul 3; Accepted 2023 Nov 19
Copyright: © 2023 Wei et al.
Copyright year: 2023
Copyright holder: Wei et al.
License: This is an open access article distributed under the terms of the Creative Commons Attribution License, which permits unrestricted use, distribution, reproduction and adaptation in any medium and for any purpose provided that it is properly attributed. For attribution, the original author(s), title, publication source (PeerJ) and either DOI or URL of the article must be cited.
License URL: https://creativecommons.org/licenses/by/4.0/

Keywords: Necroptosis, Clear cell renal cell carcinoma, BIRC3, Prognosis, Immune

Funding: The authors received no funding for this work.

==============================
Background

Necroptosis is a form of programmed cell death; it has an important role in tumorigenesis and metastasis. However, details of the regulation and function of necroptosis in clear cell renal cell carcinoma (ccRCC) remain unclear. It is necessary to explore the significance of necroptosis in ccRCC.

Methods

Necroptosis-related clusters were discerned through the application of Consensus Clustering. Based on the TCGA and GEO databases, we identified prognostic necroptosis-related genes (NRGs) with univariate COX regression analysis. The necroptosis-related model was constructed through the utilization of LASSO regression analysis, and the immune properties, tumor mutation burden, and immunotherapy characteristics of the model were assessed using multiple algorithms and datasets. Furthermore, we conducted comprehensive GO, KEGG, and GSVA analyses to probe into the functional aspects of biological pathways. To explore the expression and of hub gene (BIRC3) in different ccRCC cell types and cell lines, single-cell sequencing data was analysed and we performed Quantitative Real-time PCR to detect the expression of BIRC3 in ccRCC cell lines. Function of BIRC3 in ccRCC was assessed through Cell Counting Kit-8 (CCK8) assay (for proliferation), transwell and wound healing assays (for migration and invasion).

Results

Distinct necroptosis-related clusters exhibiting varying prognostic implications, and enrichment pathways were identified in ccRCC. A robust necroptosis-related model formulated based on the expression of six prognostic NRGs, presented substantial predictive capabilities of overall survival and was shown to be related with patients’ immune profiles, tumor mutation burden, and response to immunotherapy. Notably, the hub gene BIRC3 was markedly upregulated in both ccRCC tissues and cell lines, and showed significant correlations with immunosuppressive cells, immune checkpoints, and oncogenic pathways. Downregulation of BIRC3 demonstrated a negative regulatory effect on ccRCC cell proliferation migration and invasion.

Conclusion

The necroptosis-related model assumed a pivotal role in determining the prognosis, tumor mutation burden, immunotherapy response, and immune cell infiltration characteristics among ccRCC patients. BIRC3 exhibited significant correlations with the immunosuppressive microenvironment, which highlighted its potential for informing the design of innovative immunotherapies for ccRCC patients.

Introduction

To maintain homeostasis in both normal or stressed states, cells elicit diverse morphological and functional responses through distinct cell death pathways (Liu et al., 2022). The well-being and equilibrium of multicellular organisms heavily hinge upon programmed cell death (PCD), a process designed to eliminate cells prone to tumorigenesis or susceptible to exploitation by pathogenic bacteria for replication (Christgen, Tweedell & Kanneganti, 2022; Griffioen & Nowak-Sliwinska, 2022). Notably, necroptosis assumes a pivotal role within the framework of PCD (Yan et al., 2022). Cancer cells frequently evade from PCD by gene mutation or epigenetic modification of hub genes in the PCD pathways (Dai, Wang & Zhang, 2021; Wei et al., 2022). In contrast to prior investigations that primarily concentrated on constructing necroptosis-related models within the context of clear cell renal cell carcinoma (ccRCC) (Cai et al., 2023; Chen et al., 2022; Luo & Zhang, 2022; Qiu et al., 2022), our study transcends the realm of model development to address the critical gap in our comprehension of the underlying mechanistic processes. While these earlier publications have indeed contributed valuable prognostic models, they have not provided a comprehensive elucidation of the specific mechanisms governing necroptosis in ccRCC. Therefore, our research not only builds upon their modeling efforts but also yields novel insights into the mechanistic aspects, thereby offering a more holistic perspective on necroptosis in ccRCC.

There is growing recognition that necroptosis represents a novel form of PCD and exerts a substantial influence on the sustained proliferation and survival of certain cancer (Gong et al., 2019; Zhang et al., 2022). Necroptosis is a tightly regulated variant of necrosis reliant on the activation of receptor-interacting protein kinase 1 (RIPK1), RIPK3 and mixed lineage kinase domain-like protein (MLKL). Additionally, TNF may induce necroptosis through its regulation of the RIPK1/RIPK3/MLKL/ DRP1 axis (Al-Lamki et al., 2016; Zhao et al., 2020). Necroptosis-related genes (NRGs) have the potential to promote tumorigenesis and cancer metastasis and generate an immunosuppressive tumor microenvironment (TME) by recruiting inflammatory responses (McCormick et al., 2016; Park et al., 2009; Strilic et al., 2016). To the best of our knowledge, the TME and the tumour are intricately linked and engage in a mutual antagonistic relationship. Immune cells, inflammatory mediators, and substances secreted by tumor cells within the TME profoundly influence tumor cell growth, invasion, and metastasis. However, it is worth noting that necroptosis may also trigger a robust adaptive immune response that acts as a deterrent against tumor progression. Furthermore, it can function as a “fail-safe” mechanism when apoptosis is impaired (Feng et al., 2015; Höckendorf et al., 2016).

In the current study, we conducted a systematic investigation into the role of NRGs in ccRCC and established necroptosis-related clusters with distinct prognostic outcomes. We developed a robust predictive model centered around necroptosis-related factors, encompassing various facets related to patient prognosis and the microenvironment that surrounds the tumor. Additionally, we delved into the clinical, biological pathway, and immune characteristics of BIRC3, a pivotal gene in necroptosis.

Methods

Data acquisition and processing

We obtained the consolidated transcriptome expression matrix and clinical data of ccRCC patients from The Cancer Genome Atlas (TCGA, https://www.cancer.gov/tcga) (Koboldt et al., 2012). A list of 22 NRGs was compiled from previous research (Christgen, Tweedell & Kanneganti, 2022; Gong et al., 2019; Tang et al., 2020; Zhu et al., 2019). Additionally, we collected five datasets (GSE53757, GSE66272, GSE36895, GSE17895, and GSE73731) from the Gene Expression Omnibus (GEO, http://www.ncbi.nlm.nih.gov/geo/) database to assess the correlation between BIRC3 and clinicopathological variables in ccRCC samples. We downloaded scoring data for 526 ccRCC immunotherapy cases from the Cancer Immunome Database (TCIA; https://tcia.at/home). The clinicopathological characteristics of the patients in both the TCGA and GEO databases were listed in Table S1.

Single cell sequencing data processing

For single-cell RNA sequencing (scRNA-seq) data, we acquired three distinct datasets of ccRCC patients and normal kidney tissues from the GEO database (GSE131685, GSE152938, and GSE156632) (Liao et al., 2020; Su et al., 2021). These datasets were harmonized and merged using the “Harmony” algorithm to create a comprehensive cohort consisting of nine ccRCC and nine normal kidney tissue samples (Liao et al., 2020; Tran et al., 2020). We executed a standard Seurat workflow to unravel the inherent cellular heterogeneity in the integrated dataset. Briefly, we initially conducted principal component analysis (PCA) on the scaled data, selecting the top 26 principal components (PCs) for subsequent graph-based clustering, thereby delineating distinct cell clusters. The identification of cluster-specific marker genes was accomplished using Seurat’s “FindAllMarkers” function, with a focus on the “RNA” assay (Tran et al., 2020). Subsequently, we performed dimensionality reduction techniques, including Uniform Manifold Approximation and Projection (UMAP) and additional PCA, to further refine the cell clustering. Ultimately, cell clusters were visualized and delineated using the UMAP method. Lastly, for cell population annotation, we harnessed the “singleR” package to assign cell types to our clusters (Aran et al., 2019).

Identification of the prognostic characteristics of necroptosis-related genes

Differentially expressed NRGs between ccRCC and normal samples were identified using the Wilcoxon test and the Limma R package (P < 0.05). The “survival” package was used to explore the survival characteristics of NRGs in ccRCC. Univariate Cox regression analysis was performed for NRGs with the same expression and survival trend to further screen the prognostic NRGs.

Establishment of necroptosis-related clusters

Prognostic NRGs identified through univariate Cox analysis were used to construct necroptosis-related clusters using consensus clustering (Wilkerson & Hayes, 2010). Consensus clustering is a statistical technique used to identify robust and stable clusters within a dataset. The resulting consensus matrix provides insights into the inherent structure of the data, allowing researchers to identify groups of data points that exhibit similar characteristics. A “cluster” in this context refers to a group of data points that share similar attributes or characteristics. The overall survival (OS) for each necroptosis-related cluster was calculated by the KM curve. The log-rank test assessed survival differences (P < 0.05). Functional enrichment analysis, including Gene Ontology (GO) and Kyoto Encyclopedia of Genes and Genomes (KEGG) enrichment, was conducted to confirm cluster functions (Kanehisa & Goto, 2000; Kanehisa et al., 2016). Gene set variation analysis (GSVA) enrichment analysis was used to explore NRGs’ roles in biological pathways. Then, we downloaded the gene set of “c2.cp.kegg.v7.4.symbols” from the MSigDB database for running GSVA analysis. The gene set “c2.cp.kegg.v7.4.symbols” is a commonly used gene set from the MSigDB database, which aggregates information related to gene pathways.

Establishment of the necroptosis index

To enhance the stability and accuracy of the model, we removed four samples with a survival time of 0. Subsequently, a necroptosis-related model was constructed using NRGs significant in univariate Cox analysis via the least absolute shrinkage and selector operation (LASSO) analysis. The R package “glmnet” was employed. To assess the stability of the model, the ccRCC samples were divided into test and training sets in a 4:6 ratio. The necroptosis index (NI) formula was obtained using the linear combination of gene expression-weighted regression coefficients. The algorithm was as follows:

NI=∑k=0n⁡Coef(NRGk)∗Exp(NRGk)

where Coef was the coefficient of NRGk, and Exp was the normalized expression value of NRGk. Patients were divided into low and high NI groups based on the median NI. The model’s stability and accuracy were assessed with time-dependent receiver operating characteristic (ROC) curves, KM survival curves, and univariate and multivariate Cox regression analysis.

Identification of immune characteristics of necroptosis index

Single sample Gene Set Enrichment Analysis (ssGSEA) is a method for quantifying the enrichment of predefined gene sets within individual samples, facilitating the identification of changes in biological pathways, functions, or specific cell types using gene expression data (Hänzelmann, Castelo & Guinney, 2013). The Tumor Immune Estimation Resource (TIMER) is a tool that estimates immune cell infiltration in tumor tissues by analyzing gene expression data, providing insights into the composition of the tumor immune microenvironment (Li et al., 2017). CIBERSORT (Cell-type Identification by Estimating Relative Subsets of RNA Transcripts) is a method for inferring the proportions of different immune cell types within complex samples based on gene expression data, enabling the study of immune cell distribution in diverse biological contexts (Newman et al., 2015). Thus, ssGSEA, TIMER, and CIBERSORT were employed to assess the extent of immune cell infiltration in an individual ccRCC sample. The beeswarm package and t-test were used to analyze the differential expression of immune cells between high and low NI groups. In addition, the compare means function was used to evaluate the differences in immunotherapy scores between high and low NI groups (Table S2).

Identification of immune and prognostic characteristics of NRGs

Differential expression of NRGs in different clinicopathological stages was analyzed, the spearman correlation calculated the correlation between the NRGs and immune cells and immunosuppression checkpoints. Hub genes associated with clinical and immune features were identified. Multiple datasets were used to validate differential expression in clinicopathological variables. Next, the “h.all.v7.4.symbols.gmt” data set in GSEA was employed to investigate the biological role of the hub gene high and low expression groups in ccRCC patients. The gene set “h.all.v7.4.symbols.gmt” is a commonly used gene set from the MSigDB database, which aggregates a wide range of biological functions and pathway information.

RNA extraction, reverse transcription, and qRT-PCR

Data were collected as previously described in previous research (Han et al., 2020). Specifically, for RNA extraction, total RNAs were extracted from cultured cells or tissues using the RNA-easy isolation reagent (Vazyme, Beijing, China) following the manufacturer’s instructions. The RNA levels were assessed using RTIII All-in-One Mix with dsDNase and ChemoHS qPCR Mix (Monad, Wuhan, China). Gene expression was normalized to ACTIN, and the relative expression of mRNAs was quantified using the 2–∆∆Ct method. The primer sequences used were as follows:

BIRC3: Forward, 5′-AAGCTACCTCTCAGCCTACTTT-3′, Reverse, 5′-CCACTGTTTTCTGTACCCGGA-3′.

GAPDH: Forward, 5′-ACCCAGAAGACTGTGGATGG-3′, Reverse, 5′-TTCTAGACGGCAGGTCAGGT-3′.

Western blot assay

Data were collected as previously described in previous research (Li et al., 2023). Specifically, cells were lysed using radio immunoprecipitation assay lysis buffer (Merck KGaA; Merck, Rahway, NJ, USA), and total protein was extracted. Twenty micrograms of protein samples were separated on 10% SDS-PAGE gels, transferred onto PVDF membranes (EMD Millipore; Millipore, Burlington, MA, USA), and blocked at room temperature for 1 h. The membranes were then incubated with primary antibodies (BIRC3 concentration: 0.5 µg/mL, GAPDH dilution rate: 1:500; Abcam; Cambridge, UK) overnight at 4 °C. The following day, the membranes were incubated with a secondary antibody (dilution rate: 1:2,000; Abcam; Cambridge, UK) for 1 h at 24 °C. Signals of the target proteins were detected using an enhanced chemiluminescence detection system. The primary antibodies used in this study were as follows: rabbit polyclonal BIRC3 (24304-1-AP; Proteintech, Wuhan, China) and mouse monoclonal GAPDH (60004-1-Ig; Proteintech, Wuhan, China). These antibodies were meticulously selected for their specificity and reliability in detecting their respective protein targets.

Cell culture and cell transfection

Two human ccRCC cell lines (A498, 786-O) were procured from the cell bank of the Chinese Academy of Sciences (Shanghai, China). All cells were cultured in RPMI 1640 medium (Thermo Fisher Scientific, Inc., Waltham, MA, USA) supplemented with 10% fetal bovine serum (FBS; Thermo Fisher Scientific, Inc., Waltham, MA, USA) at 37 °C in a humidified atmosphere containing 5% CO2. In the cell culture section, cells were passaged two–three times after resuscitation before experiments to ensure their optimal state for experimentation.

Lentiviral shRNA plasmids targeting BIRC3, along with non-specific control shRNA, were obtained from Dharmacon (Shanghai, China). Transfection of plasmids and shRNA was performed using Lipo3000 following the manufacturer’s instructions. The procedure of transfection was performed as reported previously (Li et al., 2023).

Cell counting kit-8 (CCK8) assay

Briefly, A498 and 786-O cells subjected to various interventions were incubated in 96-well plates (2 × 103 cells per well) with 200 µL of culture medium and maintained at 37 °C with 5% CO2. On days 1, 2, 3, 4, and 5, 20 μL of CCK-8 solution was added to each well, followed by a 2-h incubation. Absorbance was measured at an optical density of 450 nm using a Microplate reader (Bio-Rad Laboratories, Inc., Hercules, CA, USA). The procedure of transfection was performed as reported previously (Li et al., 2023).

Transwell assay

Data were collected as previously described in Li et al. (2023). Specifically A498 and 786-O cells (with an incubation density of 2 × 105) were incubated in the upper chambers (Corning, Corning, AY, USA). For the invasion assay, the upper chambers were pre-coated with Matrigel (BD Biosciences, Franklin Lakes, NJ, USA). Culture medium without and with 10% FBS was added into the upper and lower chambers, respectively. After 12 h, non-migrated cells were wiped out while migrated or invaded CRC cells were fixed, stained and counted using an inverted microscope.

Wound-healing assay

Data were collected as previously described in Li et al. (2023). Specifically cell migration was assessed by performing a wound healing assay. Briefly, A498 and 786-O cells were transfected with BIRC3. Approximately 2 × 106 cells were seeded into six-well plates and cultured for 24 h. Then, a yellow plastic pipette tip was used to create a wound by scraping the cells. Cell migration was monitored under a Nicon Eclipse microscope and photographed at 100×.

Statistical analysis

All analyses were performed using R 4.1.0. All statistical tests were two-sided, and P-value < 0.05 was considered statistically significant unless otherwise noted. KM method and Cox regression analysis were used to analyze the prognostic characteristics of clinicopathological variables of hub gene and NI. To confirm independent factors associated with survival, univariate and multivariate Cox regression analyses were used. For our cell-based experiments, we conducted each experiment for three times to ensure the reliability and consistency of our results. Student’s t-test was performed to assess statistical differences between two groups, with analysis of variance (ANOVA) for multiple groups. Statistical significance was denoted as follows: *P < 0.05, **P < 0.01, and ***P < 0.001. These rigorous statistical analyses were employed to accurately evaluate and report the significance of our findings.

Results

Prognosis characteristics of necroptosis-related genes in ccRCC

The flow chart of this study was shown in Fig. 1. KM survival analysis identified 18 NRGs with significant connections to ccRCC patient prognosis, and 10 genes were associated with adverse outcomes, while eight showed favorable prognosis (Fig. 2). Finally, 22 NRGs were included in our study, and most of them (19/22) were differentially expressed in ccRCC tissues compared to normal tissues (Fig. 3A). Ten suitable genes were selected based on the criteria of high expression associated with poor survival or low expression associated with great survival in tumors. Univariate analysis was performed for the 10 genes, and seven genes related to prognosis were verified (Fig. 3B). The ccRCC samples were divided into four different clusters using Consensus Cluster Plus based on the expression of these seven genes in ccRCC (Figs. 3C–3E). KM curve indicated that cluster B had the best survival rate while cluster D had the worst (Fig. 3F). The heat map presented the distribution of NRGs expression profiles and clinical features (Fig. 3G). To further explore the differences in biological mechanisms between cluster B and D, we performed GO and KEGG analysis on the differential genes between cluster B and D. The biological pathway (BP) results indicated that the differential genes were related to immunity and were enriched in immune response−activating cell surface receptor signaling pathway, immune response−activating signal transduction, and T cell activation, etc. Regarding cellular components (CC), the differential genes were closely associated with cell−substrate junction, vacuolar membrane, and focal adhesion, etc. In terms of molecular functions (MF), the differential genes were mainly concentrated on GTPase regulator activity, phospholipid binding, actin binding, etc (Fig. 3H). The results of KEGG showed that the differential genes were enriched in chemokine signalling pathway, PI3K−Akt signaling pathway, MAPK signaling pathway, and NF-kappa B signalling pathway, etc (Fig. 3I). In addition, GSVA revealed the enrichment of various cancer-promoting pathways in cluster D (including the JAK STAT signaling pathway, p53 signaling pathway, and cytokine receptor interaction, etc.) (Fig. 3J). In summary, our study has revealed a close association between NRGs and the prognosis of ccRCC patients. Furthermore, we have investigated the potential roles of these genes in biological mechanisms and signaling pathways. These findings might provide valuable clues and insights for understanding the pathogenesis and treatment of ccRCC.

Figure 1 The flow chart of this study.

Figure 2 The OS Kaplan-Meier curve of 18 necroptosis-related genes in the TCGA (ccRCC samples: 539).

Figure 3 Expression and consensus cluster of necroptosis regulators in clear cell renal clear cell carcinoma.

(A) The expression difference of 22 necroptosis-related genes between normal tissue and ccRCC tissue. (B) Univariate cox regression analysis of OS for selected necroptosis-related gene with P < 0.05. (C and D) Consensus clustering model with cumulative distribution function (CDF) by k from 2 to 9. (E) Sample distribution of different clusters. (F) The KM of the overall survival in the four clusters. (G) Heatmap with gene expression and clinicopathologic characters correlation among different clusters. (H) GO enrichment analysis of different genes between cluster B and D. (I) KEGG enrichment analysis of different genes between cluster B and D. (J) Heatmap with the results of GSVA enrichment analysis among cluster B and D. Red represented activated pathways; blue represented inhibited pathways. ccRCC samples: 539, normal samples: 72. Error bars represent one standard deviation (SD) from the mean.

Construction of the necroptosis index

As shown in Fig. 4A, among the seven genes constructed for clustering, only TRAF2 showed no statistical difference between cluster B and cluster D. Lasso regression analysis identified six prognostic NRGs for the NI (Figs. 4B and 4C). Besides, the ccRCC samples were divided into the training set and test set in a 6/4 proportion. The necroptosis-related model identified six genes based on the optimal value of λ. The NI was calculated as follows: NI = 0.00482 × expBIRC3 + 0.02144 × expTNFRSF1A + 0.15676 × expTRAF5 + 0.01649 × expFASLG + (− 0.04226) × expLTAB2 + (− 0.08723) × exp TAB3.

Figure 4 Prognostic characteristics of training set and test set.

(A) Differential expression of prognostic genes between cluster B and D. (B) LASSO regression of the 6 NRGs. (C) LASSO coefficients for 6 NRGs. (D) The distribution and median value of the risk scores in the training set. (E) The distributions of OS status, OS and lactate score in the training set. (F) The distribution and median value of the risk scores in the test set. (G) The distributions of OS status, OS and lactate score in the test set. (H and I) Heatmaps of modeled gene expression in training and test sets. (J and K) ROC curves of risk model at 1, 3, and 5 years in the training and test sets. (L and M) The OS Kaplan-Meier curve of high and low NI in the training and test sets. (N and O) Univariate and multivariate COX analysis of prognostic and clinicopathological features in the training set. (P and Q) Univariate and multivariate COX analysis of prognostic and clinicopathological features in the test set. training set: 323, test set: 212. Error bars represent one SD from the mean.

Identification of the prognostic characteristics of the necroptosis index

In both the training and test sets, patients in the high-risk group had a higher mortality rate and a shorter survival time (Figs. 4D–4G). Figures 4H and 4I revealed that BIRC3, TNFRSF1A and TRAF5 were highly expressed in the high-risk group, whereas TAB2 and TAB3 were highly expressed in the low-risk group in both training and test sets. The ROC curve analysis showed that the necroptosis-related model had great predictive value for ccRCC patients’ survival in both training and test sets (training set 1-year AUC = 0.721, 3-year AUC = 0.666, 5-year AUC = 0.717; test set 1-year AUC = 0.712, 3-year AUC = 0.665, 5-year AUC = 0.621) (Figs. 4J and 4K). KM survival analysis showed that the high NI group had a lower survival rate in both the test set and the training set (Figs. 4L and 4M). In addition, univariate and multivariate cox regression analyses were utilized to further analyze the prognostic characteristics of the model. In the training set, the hazard ratio (HR) and 95% CI of NI in univariate Cox regression analysis were 1.144 and [1.065–1.229] (P < 0.001), respectively. In multivariate Cox regression analysis, HR and 95% CI of NI were 1.126 and [1.024–1.238] (P = 0.014) (Figs. 4N and 4O). In the test set, HR of the NI and 95% CI were 1.82 and [1.314–2.519] (P < 0.001) in univariate Cox regression analysis, and 1.658 and [1.131–2.432] (P = 0.01) in multivariate Cox regression analysis, respectively (Figs. 4P and 4Q). In addition, we further explored differential NI expression and survival differences in clinicopathological stages. The results showed that the NI was higher in the advanced clinicopathological stage of ccRCC patients (Figs. S1A–S1F). To investigate whether NI was applicable to patients of different clinicopathological groups, we used the KM curve to analyse whether there were prognostic contrasts between high and low NI groups among different clinicopathological groups. The KM survival curve showed that the survival prognosis of the high NI group was worse than that of the low NI group in terms of pathological stages and histological grades (Figs. S1G–S1L). Taken together, these results indicated that necroptosis-related model could be used as reliable independent prognostic factors in patients with ccRCC and could accurately predict the prognosis of the patients. Collectively, these findings suggested that the necroptosis-related model served as a reliable independent prognostic factor in ccRCC patients, could accurately predict their prognosis.

Immune characteristics of necroptosis index

In line with this, we further investigated the correlation between NI and immune characteristics, including the tumour microenvironment, immune cells, and immune checkpoints. The differences in immune cell infiltration between the high and low NI groups were analysed using CIBERSORT and TIMER. The results indicated that compared with the low NI group, Tregs cell and Macrophages were significantly higher in the high NI group (Figs. 5A and 5B). Specifically, the high NI group displayed higher immune score, stromal score and estimate score compared with the low NI group (Fig. 5C). Moreover, CTLA4 and PDCD1 were significantly higher in the high NI group than in the low NI group (Figs. 5D and 5E). Immunosuppressive cells (Myeloid-derived suppressor cells (MDSC), macrophages and regulatory T cells) were significantly over-expressed in the high NI group under the ssGSEA algorithm (Figs. 5F–5H). Additionally, we explored the difference in immunotherapy between the high and low NI groups, and the results demonstrated that CTLA4, PD1 and CTLA4+PD1 all had a great therapeutic effect on the high NI group (Fig. S2). These findings underscored the potential utility of the NI as an indicator of immune characteristics and responsiveness to immunotherapy in ccRCC patients.

Figure 5 Immune characteristics of necroptosis index.

Immune cell infiltration different expression in high and low NI groups. ((A) TIMER, (B) CIBERSORT); (C) TME score differences in high and low NI groups (including ESTIMATEScore, ImmuneScore, StromalScore); (D and E) Immune checkpoints different expression in high and low NI groups (CTLA4; PDCD1) (F–H) Immune cell infiltration different expression in high and low NI groups (MDSC; Macrophage; Regulatory.T.cell). High NI group: 294, Low NI group: 241. Error bars represent one SD from the mean.

Relationship between BIRC3 and clinicopathological variables

To investigate the clinicopathological characteristics of the six modeled genes (BIRC3, TNFRSF1A, TAB2, TAB3, TRAF5, FASLG), we further analysed differential gene expression in different clinicopathological stages. As shown in Figs. S3A–S3E, the expressions of TNFRSF1A, FASLG and TAB3 were significantly correlated with the clinicopathological variables, whereas the expressions of TAB2 and TRAF5 were not. Among the ROC curves for the six genes, BIRC3 had the highest AUC value, which was 0.948 (Fig. S3F). Correlations between six model genes and immune cells and checkpoints were then examined. The results revealed that five genes were positively correlated with tumor infiltrating immune cells and immune checkpoints with statistically significant differences. TAB3 was negatively correlated with some immune cells and immune checkpoints, but the difference was not significant (Figs. S3G and S3H). Therefore, we selected BIRC3, which exhibited the most significant immune and clinical features, for further investigation. Then, we divided the samples into high and low expression group according to the median BIRC3 expression, and further analyzed the differences in the distribution of clinicopathologic variables among different groups. The results indicated that there were significant differences in clinicopathological variables between the high and low BIRC3 groups (Grade: P = 0.03; Stage: P < 0.001; T: P < 0.001; M: P = 0.0017; N: P = 0.026), and there were more advanced patients in the high BIRC3 group than in the low BIRC3 group (Fig. 6A). In the TCGA database, the expression of BIRC3 was significantly different in different clinicopathologic stages and was higher in the advanced stages (Figs. 6B–6F). Next, we validated the clinicopathologic features of BIRC3 in the GEO datasets. As shown in Figs. 6G–6J, in the GSE53757, GSE17895, GSE36895 and GSE66272 datasets, the expression of BIRC3 in ccRCC was much higher than that in normal tissues. In the GSE73731 and GSE53757 datasets, BIRC3 was significantly correlated with pathological stages (Figs. 6K and 6L). BIRC3 was also significantly associated with histological grades (Fig. 6M) in the GSE73731 dataset. RT-qPCR was carried out in 15 pairs of ccRCC tissues and normal renal tissues and four cell lines, including three tumor cell lines and one normal renal cell line. The expression of BIRC3 in ccRCC tissues was significantly higher than that in adjacent tissues (Fig. 6N). Moreover, compared with normal cell lines, BIRC3 was significantly over-expressed in ccRCC cell lines, and the highest expression was in A498 cell line (Fig. 6O). In conclusion, BIRC3 was highly expressed in ccRCC patients and was significantly associated with clinicopathological variables. Together, our findings highlighted that BIRC3 was highly expressed in ccRCC patients and was significantly associated with various clinicopathological variables, underscoring its potential clinical relevance in the disease.

Figure 6 The relationship between BIRC3 and clinicopathological features in TCGA and GEO databases.

(A) Differences in the number of clinicopathological variables between high and low NI groups; (B–F) differential expression of BIRC3 in clinicopathological variables (such as Grade, Stage, TMN); (G–M) differential expression of BIRC3 in clinicopathological variables in GEO test datasets; (N) line chart for the relative expression of BIRC3 in ccRCC tissues (n = 15) and normal tissues (n = 15); (O) bar plot for the relative expression of BIRC3 in ccRCC cell lines and normal HK2 cell line. Biological replicates. Error bars represent one SD from the mean.

Identification of the immune characteristics and biological mechanisms of BIRC3

Figure 7A indicated that immunosuppressive cells such as MDSC, Macrophage and Regulatory.T.cell were significantly upregulated in the high BIRC3 group. And expression of BIRC3 was significantly correlated with Regulatory.T.cell, Macrophagena and MDSC, with correlation coefficients of 0.4, 0.31 and 0.46, respectively (Fig. 7B). We also analyzed the differential expression of PDCD1, CTLA4 and CD274 between high and low BIRC3 groups. The results showed that the immune checkpoints were significantly up regulated in the high BIRC3 group (Fig. 7C). Correlation analysis demonstrated that the correlation coefficients of BIRC3 with CD274, CTLA4 and PDCD1 were 0.26, 0.59 and 0.48, respectively (Fig. 7D). Besides, we further analyzed the correlation between BIRC3 and tumor microenvironment score, and the results indicated that ESTIMATEScore, StromalScore, and ImmuneScore were significantly highly expressed in the high BIRC3 group (Fig. 7E). The GSEA analysis results of BIRC3 observed that various cancer-promoting pathways were enriched in the high expression group including IL2 STAT5 signaling, IL6-JAK-STAT3 signaling, KRAS signaling, PI3K-AKT-MTOR signaling and TNFA signaling via NFKB (Fig. 7F). Finally, the volcano plot showed that BIRC3 was positively correlated with GNF-2 and PHA-665752, and negatively correlated with Rapamycin, Saracatinib, Sunitinib and Dasatinib, thus further supporting clinical treatment strategies (Fig. 7G). These findings collectively shed light on the immune characteristics and biological mechanisms associated with BIRC3, emphasizing its potential role in the tumor microenvironment and clinical therapeutic strategies.

Figure 7 Correlation between BIRC3 and immune checkpoints, immune cells, and tumor-related pathways.

(A) Differences in MDSC, Macrophage and Regulatory.T.cell between high and low BIRC3 groups, error bars represent the mean; (B) scatter plot revealing the correlation of BIRC3 and immune cells (Regulatory.T.cell Macrophage and MDSC); (C) differences in PDCD1, CTLA4 and CD274 between high and low BIRC3 groups, error bars represent the mean; (D) scatter plot revealing the correlation of BIRC3 and immune checkpoints (CD274, CTLA4 and PDCD1); (E) differences in TME scores between high and low BIRC3 groups, error bars represent one SD from the mean; (F) gene set enrichment analysis of BIRC3; (G) the volcano map showed the drugs associated with BIRC3. Red represented positive correlation; blue represented negative correlation.

In vitro functional analysis of BIRC3

This section discussed the functional analysis of BIRC3 in ccRCC cells. First, BIRC3-shRNA were transfected into A498 and 786-O cells to knock down BIRC3. Transfection efficiency was determined by RT-qPCR and Western blot (Figs. 8A–8F). We then performed CCK8 assays to measure changes in the proliferative capacity of A498 and 786-O cells. BIRC-knockdown inhibited the proliferation of A498 and 786-O cells (Figs. 8G and 8H). Transwell assay showed that BIRC-knockdown inhibited the migration and invasion of A498 and 786-O cells, with statistical significance (Figs. 8I and 8J). Wound healing assay results showed that the healing distance of A498 and 786-O cells in the BIRC-knockdown group was lower than that in the control group after 24 h, and both were statistically significant (Figs. 8K and 8L). These results indicated that BIRC3 knockdown significantly inhibited the proliferation and migration of ccRCC cells. In summary, our results provided compelling evidence that BIRC3 knockdown significantly impeded the proliferation and migration of ccRCC cells, suggesting a potential therapeutic strategy for ccRCC treatment.

Figure 8 Downregulation of BIRC3 suppressed the progression of ccRCC in vitro.

(A–F) The expression of BIRC3 was downregulated in A498 and 786-O cells, as determined by RT-qPCR and Western blot; (G and H) BIRC3-knockdown suppressed ccRCC cell proliferation in A498 and 786-O cells; (I and J) BIRC3-knockdown suppressed ccRCC cell metastasis in A498 and 786-O cells. *P < 0.05, **P < 0.01, ***P < 0.001; (K and L) wound healing assay was used to detect the effect of BIRC3-knockdown on ccRCC cell migration. Cell migration ability was represented by the wound gap distance in microscopic field at the time points of 0 and 24 h. (Biological replicates).

Single-Cell RNA sequencing analysis

To address the inherent heterogeneity of ccRCC, we employed single-cell RNA sequencing (scRNA-seq) for rigorous validation (GSE dataset). Our approach began by utilizing the “Harmony” algorithm to effectively mitigate batch effects. Subsequently, we applied the UMAP algorithm, resulting in the identification of a total of 26 distinct clusters (Fig. 9A). Our scRNA-seq analysis categorized ccRCC and normal kidney samples into primarily seven distinct cell types: epithelial cells (malignant tumor cells), endothelial cells, myeloid cells, B cells, T cells, fibroblast cells, and fibroblast-endothelial-like cells. Our investigation then delved into the distribution of necroptosis index using the single-cell signature scorer. Strikingly, we observed a significant enrichment of the necroptosis index in ccRCC tumors compared to normal tissue samples, as determined by the “AddModuleScore” algorithm (P < 0.001; Figs. 9B–9D). Furthermore, we embarked on a detailed exploration of the spatial distribution of BIRC3 within ccRCC tissues. This endeavor involved a comprehensive single-cell analysis that validated alterations in immune composition. In comparison to normal tissue samples, the expression of BIRC3 is significantly elevated in the tumor (Figs. 9E–9H). Remarkably, we noted a higher proportion of epithelial cells (malignant tumor cells) and myeloid cells in patients exhibiting elevated BIRC3 expression, particularly evident in patient samples GSM4735375, GSM4735370, and GSM4630028. On the contrary, patients with low expression of BIRC3 exhibit increased immune cell infiltration, including T cells and myeloid cells (Fig. 9I). These findings underscored the concentration of BIRC3 and necroptosis index in immune and epithelial cells, underscoring their pivotal role in modulating immune cell infiltration within the tumor microenvironment.

Figure 9 Single-cell-seq revealed the distinct landscape of necroptosis index and BIRC3 in ccRCC.

(A) “Harmony” and UMAP algorithm to remove batch effects and gathered a total of 26 clusters; (B) single-cell RNA-seq illustrated the distribution of necroptosis index by “AddModuleScore” algorithm in ccRCC; (C and D) necroptosis index wa s significantly up-regulated in ccRCC tumors tissues compared with normal samples both in all cells and tumor cells; (E and F) Single-cell RNA-seq illustrated the distribution of BIRC3 by “AddModuleScore” algorithm in ccRCC and normal tissues; (G and H) BIRC3 was significantly up-regulated in ccRCC tumors tissues compared with normal samples both in all cells and tumor cells; (I) the cell proportion among different patients and correlations of BIRC3 expression level.

Discussion

Renal cell carcinoma (RCC), which originates from the epithelial cells of the renal tubules, was a common urinary system malignancy. It accounted for 80–90% of malignant renal tumours (Scelo & Larose, 2018; Wang et al., 2022). Histologically, ccRCC was the predominant subtype and accounted for approximately 75% of RCC (Wei et al., 2021; Weng et al., 2021). While surgery is the standard treatment, drug therapies, especially targeted drugs, and immune checkpoint inhibitors, are used for advanced cases (Gulati & Vaishampayan, 2020). However, the effectiveness of currently available drugs, particularly those designed to inhibit immune checkpoints, remains variable. Further research is needed on how to predict cancer progression and treatment efficacy.

Necroptosis was programmed, caspase-independent cell death (Fulda, 2013). Unlike apoptosis, necroptosis induced inflammation through the release of damage associated molecular patterns (DAMPs). Inflammation because of necroptosis has been one of the key processes that may drive tumourigenesis and cancer progression (Ito et al., 2016; Seifert et al., 2016; Zhang et al., 2021). For example, necroptosis played an important role in promoting pancreatic ductal adenocarcinoma (PDA) progression through DAMPs or cytokines released from necrotic tumour cells, which could recruit MDSCs and tumour-associated macrophages (TAMs) to induce immunosuppressive TME (Kaczmarek, Vandenabeele & Krysko, 2013; Pasparakis & Vandenabeele, 2015). To our knowledge, immunosuppressive TME may promote tumor growth and progression (Wegner, Saleh & Degterev, 2017). In breast cancer, necroptosis was commonly found in the tumor necrosis area in advanced breast cancer tissues. By promoting the production of pro-inflammatory cytokines in tumour macrophages, it may promote the spread of breast cancer to the lung (Jiao et al., 2018). It has also been reported that necroptosis may induce chronic inflammation of the colon by activating the NF-KB pathway, thereby promoting the progression of colon cancer (Liu et al., 2015; Tortola et al., 2016). In contrast, NRGs were poorly expressed in colorectal and ovarian cancers. They were associated with poor prognosis in these patients (Bozec et al., 2016; He et al., 2013). The reason was that necrotic tumor cells could release IL-1α to activate dendritic cells (DCs) which could induce an anti-tumor immune response by producing cytotoxic IL-12 or activating CD8+ T cells (Schmidt et al., 2015; Takemura et al., 2015). Meanwhile, the tumor immune microenvironment plays an important role in the diagnosis, prevention, treatment and prognosis of ccRCC (Ke, Chen & Liu, 2022).

The prognostic characteristics and immunocorrelation of all NRGs in ccRCC were then systematically and comprehensively analysed. The results indicated that 18 NRGs were closely associated with the prognosis of ccRCC patients, among which 10 genes had negative correlation with prognosis and eight genes had positive correlation with prognosis. These differential genes were related to immunity and enriched in immune response. It was necessary to quantify the gene alteration pattern of necroptosis-related genes in each cancer, considering the individual heterogeneity of gene alteration of NRGs. Li et al. (2022) established a prognostic model that can accurately predict the survival rate of PRAD patients based on necroptosis-related genes, and the necroptosis-related model was closely related to the immune microenvironment. Zhao et al. (2021) constructed a prognostic model of necroptosis-related lncRNAs that may predict prognosis and help differentiate between hot and cold tumors, guiding individualized treatment of gastric cancer. Gu & Yuan (2022) found that miRNAs played an important role in the production and development of ccRCC and were potential biomarkers. Jiang, Chen & Chen (2020) constructed a seven-gene signature for predicting overall survival in patients with ccRCC.

In our research, we established NI as a novel marker for the evaluation of NRG alteration patterns in ccRCC. As expected, our results indicated that NI was correlated with clinicopathological characteristics of ccRCC and could be used as an independent prognostic biomarker to predict patient survival. Furthermore, we found that NI could be used to predict how patients responded to immune checkpoint inhibitors such as PD-1 and CTLA4 antibodies. To the best of our knowledge, there have been no such results published. Besides, we first systematically and comprehensively analyzed the prognostic characteristics and immune correlation of all NRGs in ccRCC. Although immune checkpoint inhibitors have been widely used in metastatic ccRCC, whether NI can be used as a marker to guide the treatment strategy remains to be investigated.

Second, we identified BIRC3 as the most immunologically and clinically relevant ccRCC NRG. BIRC3 was a member of the anti-apoptotic protein family, which may promote carcinogenesis by inhibiting cell apoptosis and promote tumor metastasis and progression through necroptosis involving multiple pathways (Chen & Huerta, 2009). For example, in hepatocellular carcinoma (HCC) BIRC3 could promote HCC epithelial-mesenchymal transition (EMT), cell migration and metastasis by upregulating MAP3K7 to induce ERK1/2 phosphorylation (Fu et al., 2019). The high expression of BIRC3 in oral squamous cell carcinoma (OSCC) was significantly correlated with lymph node metastasis, decreased survival rate, and increased cancer recurrence rate (Bhosale et al., 2017a, 2017b). In breast cancer, upregulation of BIRC3 could lead to tumor anti-apoptosis and poor prognosis (Hahm & Singh, 2013; Mendoza-Rodríguez et al., 2017; Srour et al., 2020). Our research showed that BIRC3 is an important tumour-promoting gene in ccRCC. It may be involved in the immunosuppressive tumor microenvironment. Upregulation of BIRC3 may promote cell necroptosis leading to cell expansion, plasma membrane collapse, and the release of intracellular proinflammatory factors (Interleukin 2, Interleukin 6, TNF-α and so on) (Chen et al., 2019; Saelens et al., 2005). Furthermore, the increase of pro-inflammatory factors in the TME promoted multiple inflammatory signaling pathways (such as TNF/NFKB pathway, PI3K-AKT-MTOR pathway, IL6-JAK-STAT3 pathway, IL2-STAT5 pathway, etc.), thereby recruiting many immunosuppressive cells and constructing an immunosuppressive microenvironment. Taken together, BIRC3 may promote necroptosis and regulate the expression of immune checkpoints through various biological mechanisms, thereby contributing to the establishment of an immunosuppressive tumour microenvironment.

Conclusion

To conclude, our research demonstrated that NRGs played a pivotal role in developing and prognosing ccRCC. Moreover, we have constructed necroptosis-related model in patients with ccRCC, which may be employed as a reliable predictor of prognosis and immune response. Meanwhile, BIRC3 may participate in the construction of an immunosuppressive tumor microenvironment, which may be a potential therapeutic target of ccRCC.

Supplemental Information

Supplemental Information 1 The immunotherapy score differences in high and low NI groups.

The violin plots of immunethrapy score of low- and high-NI groups. (A) CTLA4(−)+PD1(−), 95% CI: [0.03095–0.2875]; (B) CTLA4(−)+PD1(+), 95% CI: [0.1514 to 0.4516]; (C) CTLA4(+)+PD1(−), 95% CI: [0.2203–0.4815]; (D) CTLA4(+) +PD1(+); High NI group: 292, Low NI group: 234.

Click here for additional data file.

Supplemental Information 2 Validation of the prognosis of NI in the different clinicopathological characteristics.

(A) Differential expression of NI in different clinicopathological characteristics (A: cluster; B: Grade; C: Stage; D: T; E: M; F: N). (G–L)The OS Kaplan-Meier curve of NI in different clinicopathological characteristics.

Click here for additional data file.

Supplemental Information 3 Different expression of prognostic genes among clinical pathological and immune infiltration characteristics.

<!--[if !supportLists]-->(A)<!--[endif]-->Grade; (B) Stage; (C) T; (D) M; (E) N; (F) ROC of six modeled NRGs; <!--[if !supportLists]-->(G)<!--[endif]-->Correlation between genes and immune cells; (H) Correlation between genes and immune checkpoints.

Click here for additional data file.

Supplemental Information 4 Experimental raw data.

Click here for additional data file.

Supplemental Information 5 Clinical characteristics of clear cell renal cell carcinoma patients in multiple databases.

Click here for additional data file.

Supplemental Information 6 The scoring data of ccRCC immunotherapy cases.

Click here for additional data file.

Supplemental Information 7 WB of BIRC3-WB_786-O.

Click here for additional data file.

Supplemental Information 8 WB of GAPDH-WB_A498.

Click here for additional data file.

Supplemental Information 9 BIRC3-WB_A498.

Click here for additional data file.

Supplemental Information 10 GAPDH-WB_786-O.

Click here for additional data file.

Additional Information and Declarations

Competing Interests

Author Contributions

Data Availability

The authors declare that they have no competing interests.

Kai Wei analyzed the data, prepared figures and/or tables, and approved the final draft.

Xi Zhang performed the experiments, authored or reviewed drafts of the article, and approved the final draft.

Dongrong Yang conceived and designed the experiments, authored or reviewed drafts of the article, and approved the final draft.

The following information was supplied regarding data availability:

The data is available at TCGA-KIRC (https://www.cancer.gov/tcga) and at NCBI GEO: GSE53757, GSE66272, GSE36895, GSE17895, and GSE73731.

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
