# Peer review of "Identification and validation of prognostic and tumor microenvironment characteristics of necroptosis index and BIRC3 in clear cell renal cell carcinoma"

_PeerJ, doi:10.7717/peerj.16643_

## Round 0.1 · original submission · Major Revisions

Thank you for giving us the opportunity to review your work.

Consistent with the concerns and comments of the reviewers who are experts in the field, I am afraid we can not accept your manuscript in its current form. We enjoin you to review your work in line with the suggestions provided by our reviewers. All comments by our reviewers MUST be addressed and a point-by-point response provided.

We strongly advise you to have your manuscript revised by a colleague who is a proficient English language speaker or a commercial language editing and proofreading firm for clarity and logical flow.

Finally, considering that several studies have explored the prognostic value of necroptosis-related genes in clear cell renal cell carcinoma (see PMID: 36016998, PMID: 35165523, PMID: 35685693, PMID: 36910333, etc.), you may want to delineate the highlights of this particular study and your study's advantages over those similar papers. Once again, thank you for considering submission to PeerJ and we look forward to your revised work.

Reviewer 1 ·

Basic reporting

The manuscript provides an in-depth investigation into the role of necroptosis-related genes (NRGs) in ccRCC using comprehensive bioinformatics methods and laboratory experiments. The potential clinical utility of the necroptosis index (NI) in ccRCC patient prognosis, immunotherapy response, and immune cell infiltration is notable. This research also highlights the importance of BIRC3 as a possible therapeutic target in ccRCC.

Major Comments:

Abstract Clarity: The abstract seems dense with information. Consider simplifying the language and omitting non-critical details to ensure it is concise and easily understandable by a broad audience.

Methods Section:

For the data acquisition, please provide details on the patient cohort from the TCGA and GEO databases (e.g., number of samples, gender ratio, age range, and any other relevant clinical parameters).
Please specify if any ethics approval was required for using patient data.
There is an over-reliance on supplementary figures, such as Figure S1. Critical results should be in the main body, and supplementary figures should support the main text rather than being integral to understanding the study's findings, especially for a online journal.

Cellular Experiments:

Provide further details on the Western blot assay. How were the relative protein expression levels quantified, and was any normalization conducted?
In the cell culture section, details about the passage number of cells used and the source of the primary antibodies are essential.
For cell-based experiments, mention the number of replicates and whether experiments were repeated. Statistical tests used to assess significance in these assays should be specified.
Manuscript Presentation:

Minor Comments:

Terminology and Spelling: Ensure that the spelling and terminologies used are consistent throughout the manuscript, for instance, "necroptosis related genes" vs. "necroptosis-related genes."

The study's methodology seems robust, and the findings could be of clinical significance. Addressing the above concerns will help in enhancing the manuscript's clarity and impact. I look forward to the revised version.

Experimental design

no comments.

Validity of the findings

no comments.

Additional comments

no comments.

Reviewer 2 ·

Basic reporting

Please see Additional Comments.

Experimental design

Please see Additional Comments.

Validity of the findings

Please see Additional Comments.

Additional comments

The authors conducted BOTH bioinformatics analysis to work out a model (consisting of necroptosis-related genes) used for predicting prognosis of patients with clear cell renal cell carcinoma AND in vitro experiments to validate the role of one necroptosis-related gene in clear cell renal cell carcinoma. The current study would be much improved if the authors address the following concerns:

1. Necroptosis-related gene signatures in clear cell renal cell carcinoma appear to be investigated by some published papers (PMID: 36016998, PMID: 35165523, PMID: 35685693, PMID: 36910333, etc.). It would be necessary to pinpoint the research/knowledge gap, which the published papers did not fill but the current study is addressing. In other words, please summarize the novelty of this study, compared with those similar publications.

2. In all FIGURES, it would be clear and more readable to BOTH provide figures with high resolution AND expand on figure legends by explaining the meanings of colors, groups, lines, and abbreviations. These revisions would greatly help readers to understand the results and their implications easily and efficiently. For example,
2.1 In all FIGURES' bar graphs, it would be more informative to display individual data points; in other words, please replace bar graphs by EITHER scatter plots with bars OR scatter plots (a pattern like PMID: 34537192, PMID: 37046252, and PMID: 37452367). Bar graphs have been shown to be misleading, because they cannot reveal variation/dispersion within data; instead, scatter plots with bars could be acceptable and scatter plots would be preferable (as confirmed by PMID: 25901488 and PMID: 28974579).
2.2 In all FIGURES' legends, it would be more rigorous to mention BOTH the sample size (the number of data points OR how many samples/patients were included) AND whether the data points were technical or biological replicates.
2.3 In all FIGURES' legends, it would be more rigorous to mention how the authors reported the data error (variation/dispersion): standard deviation (SD), confidence intervals (CI), or standard error of the mean (SEM, which would be not preferable).

3. In ABSTRACT:
3.1 In Methods ("We conducted multiple bioinformatics analyses, including prognostic analysis and cluster analysis, using R software"), it would be more informative and rigorous to expand on this sentence. The current version did not seem to provide any useful information about the methods used for all analyses prior to BIRC3-related experiments. In other words, the authors did not seem to do a good job of summarizing their (a little bit) complicated bioinformatics analyses; this paucity of summaries prevented readers from understanding this manuscript. Thus, it would be clearer (easier to understand) to add a figure showing the workflow of this study.
3.2 If the comment 3.1 cannot be addressed, in the current manuscript, readers would be confused by some words that were not mentioned in Methods but appeared in Results: "Necroptosis related clusters", "different prognostic and enrichment pathways", "necroptosis index (NI)", "prognostic NRGs", and "immune characteristics, tumor mutation burden and immunotherapy response". Thus, please rewrite Methods and Results; otherwise, it would be very difficult for readers to understand Results.
3.3 Similarly, in Conclusion, "Necroptosis prognosis model" was an expression that has never appeared in prior sections, readers would be confused by this unclear expression. If this term refers to the "necroptosis index (NI)", please use only one expression throughout the manuscript. Furthermore, using "model" could be better than saying "index".

4. In INTRODUCTION:
4.1 In Paragraph 1 ("However, the specific mechanism of necroptosis in clear cell renal cell carcinoma (ccRCC) remains unclear"), it would be more accurate and rigorous to rewrite this sentence after addressing the comment 1 and thinking over specific advantages of the current study over all other similar publications (PMID: 36016998, PMID: 35165523, PMID: 35685693, PMID: 36910333, etc.).
4.2 In Paragraph 3 ("We constructed stable prognostic models representing different prognostic and microenvironmental characteristics"), it would be clearer (easier to understand) to rewrite this sentence by replacing "prognostic and microenvironmental characteristics" with another expression. The idea that "models have prognostic/microenvironmental characteristics" seems very confusing. If this type of expression is widely used in this field, please cite references to showcase & explain the meaning of this expression.

5. In METHODS:
5.1 It would be more rigorous to add a paragraph introducing how the authors did RT-qPCR and what primers (with sequences) were used.
5.2 In "Data acquisition and processing" ("The GSE53757, GSE66272, GSE36895, GSE17895, and GSE73731 datasets were obtained ..."), it would be more rigorous to briefly mention basic information about these datasets (like how many normal & tumor samples were included, etc.). Ideally, please add a table for this information (a pattern like the Table 1 in PMID: 331780396).
5.3 In "Data acquisition and processing" ("The Cancer Immunome Database (TCIA) was used to download the scoring data of 533 ccRCC immunotherapy cases"), it would be more informative and clearer to expand on "the scoring data of 533 ccRCC immunotherapy cases". Otherwise, readers could be confused by why the authors selected these "533 ccRCC immunotherapy cases" and what is meant by a "scoring data".
5.4 In "Establishment of necroptosis related clusters" ("Based on the expression profiles of significant NRGs analyzed by univariate cox analysis, necroptosis related clusters were constructed using Consensus Clustering"), it would be more informative and clearer to explain what is "Consensus Clustering", how "Consensus Clustering" works, and what is meant by "cluster". Otherwise, it could be very difficult for readers (especially those without expertise in this skill) to understand all results about "clusters".
5.5 In "Establishment of necroptosis related clusters" ("we downloaded the gene set of "c2.cp.kegg.v7.4.symbols" from the MSigDB database for running GSVA analysis"), it would be more informative and clearer to briefly introduce the gene set "c2.cp.kegg.v7.4.symbols".
5.6 In "Identification of immune characteristics of necroptosis index" ("The ssGSEA, TIMER and CIBERSORT were used to determine the relative level of immune cell infiltration in a single ccRCC sample"), it would be clearer (easier to understand) to explain why these three tools were utilized and cite their references.
5.7 In "Identification of immune and prognostic characteristics of necroptosis related genes" ("the “h.all.v7.4.symbols.gmt” data set in GSEA was employed to investigate the biological role of the hub gene high and low expression groups in ccRCC patients"), it would be more informative and clearer to briefly introduce the gene set "h.all.v7.4.symbols.gmt".

4. In RESULTS:
4.1 It would be clearer to end each paragraph in RESULTS with one sentence: "Together, these results suggest that ..." (a pattern like PMID: 37452367, PMID: 34715879, PMID: 34384362, PMID: 35965679, and PMID: 34537192), summarizing a paragraph AND highlighting the implications of all results in the paragraph.
4.2 In "Prognosis characteristics of necroptosis related genes in ccRCC" ("Ten suitable genes were screened according to the criteria of high expression + poor survival or low expression + great survival in tumors"), it would be clearer to rewrite this sentence by clarifying whether the "ten suitable genes" were those in "10 genes were negatively correlated with prognosis .... (Figure S1)" or any other 10 genes.

---

## Round 0.2 · Minor Revisions

Thank you for taking time to address most of the issues raised by our expert reviewers, however, one of our reviewers notes that some issues have not been addressed thoroughly. Please kindly pay attention to these comments below:

1. As to my previous comment 2.1 (In all FIGURES' bar graphs, it would be more informative to display individual data points; in other words, please replace bar graphs by EITHER scatter plots with bars OR scatter plots (a pattern like PMID: 34537192, PMID: 37046252, and PMID: 37452367). Bar graphs have been shown to be misleading, because they cannot reveal variation/dispersion within data; instead, scatter plots with bars could be acceptable and scatter plots would be preferable (as confirmed by PMID: 25901488 and PMID: 28974579)), the authors have not displayed the individual data points in the bar graphs of Figure 6O, Figures 8A, 8C, 8D, 8F, and 8I–L. This revision would make the study more rigorous.

2. As to my previous comment 2.2 (In all FIGURES' legends, it would be more rigorous to mention BOTH the sample size (the number of data points OR how many samples/patients were included) AND whether the data points were technical or biological replicates), the authors have not mentioned the sample size for Figures 6B–M, Figures 7A–E, Figures 8A, 8C, 8D, 8F, 8I–L, Figures 9C–D, and 9G–H.

Reviewer 1 ·

Basic reporting

The authors have addressed all my concerns.

Experimental design

No comments.

Validity of the findings

No comments.

Additional comments

No comments.

Reviewer 2 ·

Basic reporting

Thank the authors for responding to the comments. However, some issues could not have been addressed thoroughly:

1. As to my previous comment 2.1 (In all FIGURES' bar graphs, it would be more informative to display individual data points; in other words, please replace bar graphs by EITHER scatter plots with bars OR scatter plots (a pattern like PMID: 34537192, PMID: 37046252, and PMID: 37452367). Bar graphs have been shown to be misleading, because they cannot reveal variation/dispersion within data; instead, scatter plots with bars could be acceptable and scatter plots would be preferable (as confirmed by PMID: 25901488 and PMID: 28974579)), the authors have not displayed the individual data points in the bar graphs of Figure 6O, Figures 8A, 8C, 8D, 8F, and 8I–L. This revision would make the study more rigorous.

2. As to my previous comment 2.2 (In all FIGURES' legends, it would be more rigorous to mention BOTH the sample size (the number of data points OR how many samples/patients were included) AND whether the data points were technical or biological replicates), the authors have not mentioned the sample size for Figures 6B–M, Figures 7A–E, Figures 8A, 8C, 8D, 8F, 8I–L, Figures 9C–D, and 9G–H.

Experimental design

N/A

Validity of the findings

N/A

Additional comments

N/A

---

## Round 0.3 · accepted · Accept

Congratulations! Thank you for addressing all review comments and making use of all the suggestions provided by our expert reviewers to improve the quality of your paper.